# The Relation among Organizational Culture, Knowledge Management, and Innovation Capability: Its Implication for Open Innovation

**Long Lam** [1,2], **Phuong Nguyen** [3,*] , **Nga Le** [3] **and Khoa Tran** [2]

1   Board of Management, Quang Trung Software City, Ho Chi Minh City 700000, Vietnam; long@qtsc.com.vn
2   The School of Business, International University, Vietnam National University,
    Ho Chi Minh City 700000, Vietnam; ttkhoa@hcmiu.edu.vn
3   Center for Public Administration, International University, Vietnam National University,
    Ho Chi Minh City 700000, Vietnam; lttnga@hcmiu.edu.vn
*   Correspondence: nvphuong@hcmiu.edu.vn

**Abstract:** It is widely acknowledged that enhancing innovation capability is an inevitable requirement for the survival and sustainable growth of firms operating in the information technology sector. Therefore, this study was conducted to explore the relationship among organizational culture, knowledge management and innovation capability in the open innovation environment to provide useful suggestions and recommendations for managerial practices within the high-tech industry. Primary data collected from 182 high-tech firm's representatives were processed by using the Structural Equation Modeling approach. The results showed that knowledge management was strongly correlated with innovation capability. The positively significant relationship between organizational culture and knowledge management was also confirmed. Overall, the findings suggest that an open innovation culture of an organization in which mutual trust, collaboration and learning are promoted by supportive and participative leaders is more likely to increase the efficiency of knowledge management practices; thus, eventually lead to enhanced innovation capability of the firm.

**Keywords:** organizational culture; knowledge management; innovation capability; open innovation environment

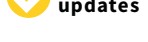



## 1. Introduction

The information technology (IT) industry in Vietnam has grown rapidly in recent years, contributing a significant proportion of market share to the growth of the economy. Especially after the introduction of digital transformation, the IT sector quickly became one of the most attractive industries in Vietnam. It is noticeable that the high-tech market gains favorable evaluations from not only domestic investors but also foreign investors, evidenced by the fact that, in 2018, this industry received 2.1 billion USD of foreign direct investment. However, such fast transitions and development have created intensive competition in the industry, which can be considered as an exploitable opportunity as well as a major challenge for IT firms' growth and survival.

In order to keep up with the trend of widespread globalization, rapid changes in open innovation and production technologies, high-tech companies are gradually changing development strategies [1]. Managers nowadays do not concentrate only on creating values in the number of plants, equipment and products but also the intellectual property, customer service, capability of collaborating with partners, telecommunication infrastructure, and the creativity and potential skill of employees [2,3]. In other words, firms no longer seek competency and sustainable growth in allocating existing limited natural resources but in innovating and creating intellectual assets [4]. This is especially crucial in the case of organizations in the IT sector as improving innovation capability is an inevitable requirement for survival and long-term growth.

Open innovation is considered as a novel open business model that can inspire diversified knowledge to be turned into creative results [5–7]. In other words, it provides mechanisms for organizations to exploit inflows and outflows of knowledge to become more creative [8]. Firms can adopt open innovation to improve their internal innovation process by utilizing external resources [9]. Innovation becomes increasingly important in creating and maintaining an organization's competitive advantages as well as contributing to its growth and prosperity [10,11]. Several recent studies have illustrated that firms possessing strong innovative capability can timely capture market opportunities and thus, proactively respond to external changes and customer demands [10,12,13]. Achieving this objective depends greatly on the manager's ability to identify innovation influential factors. A knowledge-nurtured business culture and efficient knowledge management practices play critical roles in enhancing the innovation capability of an organization. Thus, in this study, the importance of culture, knowledge processes and innovation capability is demonstrated in order to provide useful insights and managerial implications for effective operational development.

This research is conducted with the aim to explore the relationship among organizational culture, knowledge management and innovation capability under the context of IT firms located in Science Parks and High Technology zones in the South of Vietnam. Specifically, the research is designed to answer the following questions:

1. Under the context of IT firms in Vietnam Science parks, whether do leadership styles attribute to the formation of a strong organizational culture?
2. How does organizational culture affect knowledge management processes in IT firms in Vietnam High Technology Zones?
3. Whether there is a direct relationship between knowledge management and innovation capability of IT firms in Vietnam High Technology Zones?

The initial purpose of this research is to determine whether supportive and participative leadership behaviors can contribute to the formation of a solid organizational culture in Vietnam high-tech firms. As Vietnam is attempting to develop a more knowledge-based and technology-driven economy, the characteristics of organizational culture that could sustain such an economy need to be established.

Secondly, the study aims to explore the relationship between organizational culture and knowledge management implementation from Vietnamese IT firms. There is a general agreement among scholars and practitioners that a knowledge-sharing supportive organizational culture is essential to develop knowledge management incentives [14]. How information and knowledge, created within and outside the organization is used depends on the structural, socio-psychological and geographical set up of an organization [15]. However, few authors have examined the contribution of organizational culture to the knowledge management process or considered culture as a knowledge resource. This study aims to fill the gap and attempts to demonstrate that culture can be a crucial factor when firms attempt to apply an efficient knowledge management system.

Thirdly, the study investigates whether open innovation knowledge management has any impact on innovation capability. Overall, the academic research on the attributes and consequences of effective knowledge management still remains unexplored [3]. Regardless, there is a general agreement among several researchers that knowledge management can significantly contribute to the improvement of an organization's innovation capability. For instance, research by [16] comments that effective management of knowledge results from strong internal collaboration increases the opportunity for the employees to capture the prior information and engage in creative activities more enthusiastically. As innovation depends heavily on the availability of knowledge, the richness of information in the modern business environment has increased the complexity and uncertainty in the innovation process. Thus, it is of major importance for firms to develop effective ways to manage knowledge assets [17]. To our knowledge, there has been limited studies conducted to investigate the mentioned relationship in the case of Vietnam Science parks and High Technology zones. Thus, by focusing on a specific group sample under the fast-changing

industry, we aim to discuss and identify whether companies and organizations operating in high technology parks in Vietnam can improve technology capability while exploiting an efficient knowledge management strategy.

## 2. Theoretical Framework and Hypotheses

### 2.1. Leadership Styles and Organizational Culture

Leadership style is considered to be one of the most crucial organizational elements in order for firms to compete successfully and gain sustainable advantages. Leadership style is defined as an influential relationship between leaders and followers that results in the achievement of their shared purposes [18]. The rich knowledge from the related extant literature provides numerous dimensions of leadership that are widely accepted among scholars and practitioners. A recent study by [19] developed the Path-Goal theory, which has been receiving more attention from researchers in this stream of literature due to its applicability in management practices trend nowadays [18]. The Path-Goal theory describes several managers' tasks, these include identifying employees' roles and responsibilities, setting benchmarks for success, providing guidance and coaching, removing obstacles that can prevent task completion. Importantly, the theory also suggests that a manager should use different leadership styles available including supportive, participative, achievement-oriented and directive [19].

In this research, two dimensions of leadership as proposed in Path-Goal theory: supportive and participative are adopted to measure for the leadership styles variable. Supportive leaders are managers who provide emotional and instructional support for subordinates. They tend to show high concern for their followers' well-being and take account of followers' needs and preferences when making decisions. A participative leader is generally referred to as someone who regards employees' opinions from different levels during the decision-making process. [16,19].

Culture has inherently been an essential driver of innovation. It enables firms to control open innovation complexity [5,20]. Open innovation implies the development of new values that have been generated by integrating the markets and innovations of various businesses outside organization borders, and the implementation of new and combined business models [21–26]. In a recent study, Yun et al. [20] consider culture for open innovation as complexity and dynamics. It is built on "values such as curiosity, creativity, flexibility, and diversity, because the open dimension requires values such as openness, trust, responsibility, authenticity, and sustainability." However, in this study, organizational culture is adopted a less dynamic concept. Organizational culture is defined as a firm's internal characteristics that play a determinant role in its long-term development. It represents how organization members interact with one another and how the organization associates with its stakeholders. In other words, a business' culture is a guidance that directs the operation, workflow and customer management within an organization [15,27]. The dimensions and attributes of organizational culture have been studied excessively by scholars under various contexts [14]. Moreover, organizational culture comprises implicit and unwritten rules and that employees are expected to be aware of to apply in daily work routine [28]. As mentioned in research by [29], organizational culture is defined as a design of fundamental presumptions created by the labor community to develop an integrative system dealing with external factors as well as coordinating internal relationships within an organization. Another study from [30] characterizes organizational culture as a shared cognitive system which acts as a guidance for perceptions, thoughts and language of organization members.

The relationship between organizational culture and leadership styles in the open innovation environment has been a research topic that receives intensive investment from organizations and institutions over the world [31]. Despite several references in academic and literature have confirmed the connection between leadership and corporate culture, there has been limited comprehensive study carried out to explore the specific characteristic of this relationship [32]. Thus, the fact that there is a controversial conclusion regarding

their relationship is undeniable. Many researchers stated that organizational culture plays a crucial role in the development of effective leadership [33]. For example, [34] argue that the cultural values, trends and rules within an organization significantly affect the management style of leaders. In a study by [35], the authors confirm that business culture is a key factor in forming effective leadership styles in an organization. In another perspective, the article from [28] claim that leaders are the ones to set the norms, beliefs, policies and procedures of an organization during the initial stage of business creation. However, as the firm matures, the strategic culture and its characteristics are the determinants of leadership styles.

In another study, Schneckenberg [36] stresses the value of internal incentives and the leadership role in building an internal culture that will contribute to the successful encouragement of open innovation practices. He shows that monetary rewards are not inherently the best approach to gain open innovation adoption, management-driven such as immaterial and task-content incentives can have more optimal outcomes. Leadership role plays an essential role in enhancing culture changes and stimulating employees to contribute new ideas to integrate the open innovation environment [20].

On the other hand, another school of research focuses on the role of leadership styles on the development of organizational culture. As stated by [37], leaders have tremendous flexibility to determine how their organizations will be managed, and thus, have a significant effect on the culture of an organization. The author's finding from exploiting data of employees working in an international port city in China also confirms that leadership is a strong predictor of business culture. In addition, the study by [31] recognizes that leadership becomes a factor of organizational culture and is incorporated into the daily organizational routine. Findings from [38] imply that the management style of leaders can significantly affect the employees' perception of the cultural values of the organization. Therefore, following more recent researches, which supports the statement that leadership is the determinant of business culture [31,37], the following hypothesis is proposed:

**Hypothesis 1 (H1).** *Leadership styles have a positive relationship with organizational culture.*

*2.2. Organizational Culture and Knowledge Management in the Open Innovation Environment*

As mentioned in the above section, organizational culture is referred to as a set of norms, procedures, beliefs and core values that guide and direct its members' thinking and behaviors toward each other as well as the organization's related stakeholders [39]. An examination of the definitions and conceptualizations of organizational culture by different researchers uncovers the complexity and assortment of this factor. Previous studies have been conducted to evaluate some of the business culture's behavioral components. However, the valid and reliable instruments used to measure the important and recognizable characteristics of culture under high technology firm context still remain limited. Especially research for IT firms locating in industrial zones in the South of Vietnam is even more unexplored. In addition, major businesses in industrial zones create and retain dominant designs and technical regimes by open experimentation with start-ups, small and medium-sized companies and social enterprises [22].

In order to timely react to rapid changes of market and customer demand, firms are encouraged to develop a knowledge-nurtured culture. It is widely recognized that knowledge is not only an important resource of a firm but it also serves as a basic source of competitive advantages [4,12]. An efficient knowledge management strategy allows organizations to prepare and overcome environmental challenges and changes [40]. As information administration is gaining more importance in modern managerial practices, many practitioners and scholars have conducted extensive studies to identify its dimensions. Basically, knowledge management includes practices of identification, acquisition, creation, storage, sharing and use of knowledge by individuals and groups within an organization [3,11,41]. In terms of the organizational capabilities perspectives, knowledge management is composed of technology, structure and culture along with knowledge process architecture of acquisition,

conversion, and protection [42]. Those factors are considered essential for an organization's preconditions for an effective knowledge management strategy. As mentioned by [43] knowledge management strategies consist of three interrelated processes: knowledge acquisition, knowledge conversion and knowledge application.

It has been confirmed that organizational culture plays an important role in developing knowledge management. How firms interact with related stakeholders determines the efficiency of managing external information, which in turn, affects the firms' ability to implement open innovation [44]. Some previous studies have found that organizational culture is the foundation of knowledge initiatives as it can encourage members to learn and share new information [30,35]. As argued by [45] organizational culture is a key factor in facilitating an effective knowledge management process, including knowledge creation, transfer, and application of new and existing knowledge. In research examining the relationship between these two factors exploiting data from telecommunication firms, [4] find that a culture in which mutual trust, collaboration and learning are promoted is significantly related to effective knowledge management. In a study by [46], organizational culture is found to have the strongest impact on knowledge management among other factors. The findings from a recent study by [18] also confirm the positive correlation between the two variables. The research of [14] finds that organizational culture also contributes significantly in the success of knowledge management implementation. Following the mentioned studies, the below hypothesis is proposed:

**Hypothesis 2 (H2).** *Organizational culture has a positive relationship with knowledge management.*

### 2.3. Leadership Styles and Knowledge Management

Besides organizational culture, leadership styles also determine the effectiveness of knowledge management implementation. In order to harvest the competitive advantages through information administration practices, managers' supportive attitude and human-centered appreciation mindset can encourage employees to learn and adapt new and existing knowledge enthusiastically [16]. It can be said that, closed-minded labor force, as well as management styles, can be a barrier to knowledge management, which can lead to a reduction in the innovation capability of the whole organization. There have been various researches conducted to examine leadership styles using the measurement for these two dimensions. For instance, a research by [47] conducts a survey of 227 respondents working in the autonomy industry and finds that supportive and considerate leader behaviors strongly promote information and knowledge sharing among managers and subordinates. As mentioned in a study by [2], human factors are essential in the process of nurturing knowledge-based. In a study using the database of 2703 firms in Germany, participative and supportive behaviors encourage cooperative and collaborative bonds among leaders and followers in the implementation of knowledge management practices as a result of improving communication [16]. The study from [48] also stresses the importance of the aforementioned management styles in accelerating information and knowledge sharing within an organization. Similarly, the findings from [18] confirm the positive relationship between leadership styles and knowledge management. Based on these arguments, we also come up with the following hypothesis:

**Hypothesis 3 (H3).** *Leadership styles have a positive relationship with knowledge management.*

### 2.4. Knowledge Management and Innovation Capability

In today's competitive business environment, innovation is a crucial factor for firms' survival and development [49]. Innovation capability is a topic that receives tremendous attention from many researchers, thus the extant literature has provided a wide range of definitions and dimensions for this factor. For instance, a study from [50] states that innovativeness is a multi-dimensional construct that incorporates the intention to be innovative, the framework to foster development, essential operational behaviors to influence a market

and value orientation, and the environment to actualize innovative advancement. High innovation capability allows firms to generate fundamental values and beliefs that guide employees to convert knowledge into new intellectual assets, such as improvement of existing products, services, processes, technology, and administrative systems, which in turn, secure the long-term survival and sustainable development of the organizations [51,52]. In a qualitative research on a case study by [53], the author confirms that even though open innovation strategy is an important facilitator of entrepreneurial performance, firms pursuing the aforementioned strategy have to face major difficulties, such as internal and external conflicts, competitions. Thus, innovation capability helps firms to deal with the complexity as well as the emergence of a business environment.

Organizational innovativeness is closely related to creating and exploiting knowledge resources available within organizations. To be more specific, knowledge management can play an important role in supporting and nurturing innovation [10,54]. A qualitative research from [55] on 78 Spanish small and medium-sized enterprises (SMEs) confirms that the flow of information on business demands and technology potential, as well as networking with customers and partners, is a concrete precondition for initiating innovation within firms. Thus, it can be said that efficient knowledge management can contribute to the enhancement of business's competitive advantages, customer focus, employee relations and development, innovation and reducing costs. Various scholars have proposed the importance of knowledge administration within the organizations, hence, implying that the implementation of the aforementioned practices would be conductive process innovation [51,55,56]. By designing and implementing a system of knowledge sharing, firms are forced to make changes in the traditional operation mindset concerning managing intellectual property, and employee working styles by adopting new processes, disciplines and cultures, as result of constituting organizational innovation. Thus, the following hypothesis is proposed:

**Hypothesis 4 (H4).** *Knowledge management has a positive relationship with innovation capability.*

The conceptual model was built based on hypotheses developed from the extant literatures as mentioned in the above section. The model is presented in Figure 1 below.

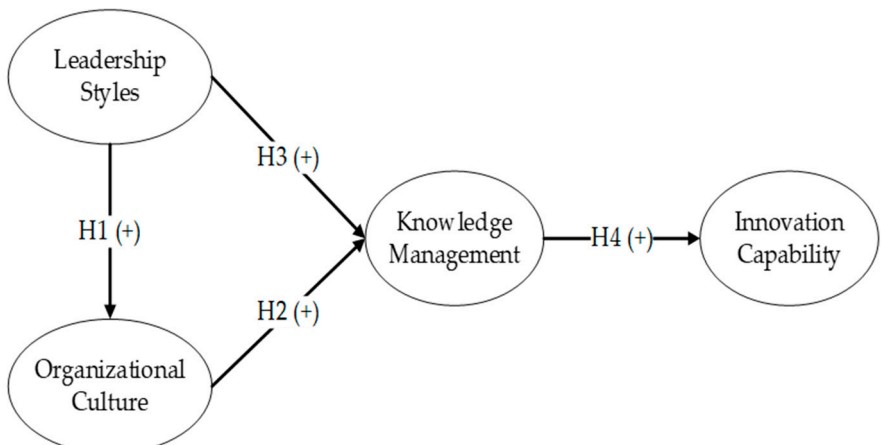

**Figure 1.** Research model.

## 3. Methodology

### 3.1. Statistical Analysis Techniques

In this research, we conducted a quantitative approach to examine the relationships among cultural factors, knowledge management and innovation capability of Vietnamese IT firms located in High Technology Zones, namely Quang Trung Software City and High Technology Zone in the South of Vietnam. Since the data used in this study is primary data

collected with the purpose of enriching the literature in the business management field, we conducted the survey research via the use of questionnaires.

After completing the data collection, the results were presented by using SmartPLS3 software to conduct Confirmatory Factor Analysis and Structural Equation Model (SEM) analysis. SEM is a statistical technique for evaluating causal effect relationships of the intended variables. In addition, the partial least squares (PLS-SEM) approach is highly appropriate for analyzing a complex research model with multiple simultaneous relationships among variables, which is common in the business management research field [57]. In addition, since managers are the representative sample of this research, the use of PLS-SEM is obviously an advantage [58].

In order to apply Structural Equation Model (SEM) for data analysis, the optimal sample size should be as ten times as the tested indicators [59,60]. Whereas, the acceptable minimum population is at least five times as many observations as the number of variables to be analyzed. In our study, there is a total of 19 indicators, meaning that the ideal sample size needed for this research is 190. However, following the standard rule, the minimum sample size of 95 is also considered acceptable.

### 3.2. Data Collection

A survey integrating current scales from published high-quality academic articles was designed to investigate the interactions between the variables as proposed in the research model. The intended participants in this study included team leaders and managers from different levels of IT firms located in Vietnamese Science parks, namely, Quang Trung Software City (QTSC) and Saigon Hi-Tech Park (SHTP). The authors were officially given access to obtain information from a representative sample of managers within the aforementioned organizations. Initially, an estimated sample of 190 IT firms was to be collected to examine the proposed hypotheses. However, the actual number of responses we could collect was 182 firms. Since the intended participants of this research were managers at different levels of IT firms, it is difficult for the authors to obtain the desired sample size of 190 as initially planned within the limited time frame. Regardless, the sample size of 182 is considered close to the optimal size, and lies within the acceptable sample range, in accordance with the standard rule by [59,60].

The questionnaire contained items measuring the scales of latent variables as well as demographic characteristics of the respondents. The dimensions used for measuring the constructs postulated in this study had been specifically checked in the existing literature and verified to be valid and reliable measures. Before sending the questionnaire to the respondents, it was translated to Vietnamese under the revisions of QTSC partners, who have deep knowledge about the IT sector. After reviewing several times via online and offline meetings with QTSC partners to identify any remaining ambiguities or misleading statements with the instruments as well as make any amendments where necessary, the final version of the questionnaire was sent out to the intended respondents via Google form. We consider online surveys as the most appropriate method to collect data at the moment, as during the time that this research was conducted, the situation of COVID-19 became quite severe. Furthermore, online surveys allow us to save from printing, postage costs, the time required for data entry and receiving/processing data was also significantly reduced.

The questionnaire sent to the intended participants included the information about our research group, the purpose of our research project and assured them that their responses will be used for academic purposes only. The questionnaire only asked for the information of the business that they are working for, such as business types, years of operations, etc. . . . , as such data are important to make meaningful conclusions and suggestions for our implication and interpretation.

Table 1 depicts the characteristics of the sample company representatives that participated in the survey. To achieve a proper generality of the research findings, we surveyed participants from IT firms, specifically, business types and sizes, job title of the respondents as well as organizational lifecycle phases were asked for demographic information. It is

worth mentioning that the respondents who work as managers and team leaders account for more than 30% of the population, the remaining include 20% of board of director members, and 6.6% of the administrative council. The reason our analysis focuses on informants who are at the management level is that they are the company's most significant individuals who have key knowledge and understanding of how their business operates.

**Table 1.** Demographic characteristics (*N* = 182).

| Items | Description | Sample | Percentage (%) |
|---|---|---|---|
| Type of business | Limited liability company | 75 | 41.2 |
| | Joint stock company | 57 | 31.3 |
| | Partnership | 2 | 1.1 |
| | Joint venture company | 2 | 1.1 |
| | Private enterprise | 4 | 2.2 |
| | Individual business households | 3 | 1.6 |
| | State-owned companies | 16 | 8.8 |
| | Companies with foreign capital | 21 | 11.5 |
| | Others | 2 | 1.1 |
| Job title | Administrative council | 12 | 6.6 |
| | Board of directors | 38 | 20.9 |
| | Manager | 65 | 35.7 |
| | Team leader | 67 | 36.8 |
| Number of employees | Less than 20 | 49 | 26.9 |
| | 20–50 | 54 | 29.7 |
| | 51–99 | 13 | 7.1 |
| | 100–200 | 38 | 20.9 |
| | More than 200 | 28 | 15.4 |
| Years of operations | Less than 3 | 32 | 17.6 |
| | 3–5 | 49 | 26.9 |
| | 6–10 | 27 | 14.8 |
| | 11–20 | 56 | 30.8 |
| | More than 20 | 18 | 9.9 |

*3.3. Measurement*

In order to measure the latent variables proposed in the literature reviews, proposed items were adopted instruments used from previous studies, each item is measured by using five-point Likert scales (1 = strongly disagree to 5 = strongly agree). The items had been modified slightly when translated to Vietnamese in order to match with the domestic context and prevent targeted respondents from misreading and ambiguities.

Firstly, the items used for measuring leadership styles (LEA) were adopted from [48]. In their study, leadership behaviors were divided into participative and supportive styles which are believed to be suitable under the modern context of IT firms. As the business environment nowadays encourages the appreciation of human capital, managers who show high concern and value their subordinates' welfare and opinions are more likely to create a stronger connection culture within the organization. Therefore, we selected and modified five items to measure LEA from the mentioned study, including before making decisions, They considers what his/her subordinates have to say (LEA1), They helps people to make working on their tasks more pleasant (LEA2), They looks out for the personal welfare of group members (LEA3), They treats all group members as equals (LEA4), They schedules the work to be done (LEA5)

Six items from [4] were used to measure organizational culture. These items were developed to estimate the dimension culture of knowledge incentives within an organization, including collaboration, mutual trust and learning as presented in Table 2.

**Table 2.** Questionnaire constructs and variables.

| Constructs | Items | Observed Variables |
|---|---|---|
| Leadership Styles (LEA) | LEA1 | Before making decisions, they considers what his/her subordinates have to say. |
| | LEA2 | They helps group members to make working on their tasks more pleasant. |
| | LEA3 | They looks out for the personal welfare of group members. |
| | LEA4 | They treats all group members as equals. |
| | LEA5 | They schedules the work to be done. |
| Organizational Culture (OC) | OC1 | Our organization members are satisfied by the degree of collaboration. |
| | OC2 | There is a willingness to collaborate across organizational units within our organization. |
| | OC3 | Our company members have reciprocal faith in others' ability. |
| | OC4 | Our company members have reciprocal faith in others' behaviors to work toward organizational goals. |
| | OC5 | Our company provides various formal training programs for the performance of duties. |
| | OC6 | Our company encourages people to attend seminars, symposia, etc. |
| Knowledge Management (KM) | KM1 | Our company creates new knowledge for application across functional boundaries. |
| | KM2 | Our company creates operations systems for application across functional boundaries. |
| | KM3 | Our company has a standardized reward system for sharing knowledge. |
| | KM4 | Our company engages in processes of integrating different sources of knowledge across functional boundaries. |
| Innovation capability (INC) | INC1 | Our company develops new production methods and procedures. |
| | INC2 | Our company introduces newer (or improved) management methods and procedures than three years ago. |
| | INC3 | Our company introduces newer (or improved) products than three years ago. |
| | INC4 | Our company modifies and/or improves existing products. |

Thirdly, we adopted four items used in the research of [61] to measure knowledge management (KM). They included items such as: Our company creates new knowledge for application across functional boundaries (KM1), Our company creates operational systems for application across functional boundaries (KM2), Our company has a standardized reward system for sharing knowledge (KM3), Our company engages in process of integrating different sources of knowledge across functional boundaries (KM4). Finally, six items were adopted from [62] to explore innovation capability. The specific items can be seen in Table 2 below.

## 4. Results and Discussions

### 4.1. Testing for Convergent and Discriminant Validity

The scales measuring the constructs in this article were tested for construct validity. A measure can be said to have construct validity if it can test the hypothetical construct or characteristic as its design [57]. To test for convergent validity of the construct, we examined Cronbach's alpha, average variance extracted (AVE), and composite reliability (CR). As recommended from previous studies, the acceptable threshold values for Cronbach's alpha, AVE and CR are 0.6, 0.5 and 0.7, respectively [57,63].

As can be seen from Table 3, all results of the Cronbach's alpha were higher than 0.8, which exceed the acceptable criteria. The Average variance extracted (AVE) values were all above 0.5, ensuring the convergent validity of the tested constructs. Additionally, the factors' composite reliability was also above 0.7, indicating high internal consistency [57]. Furthermore, the fact that all the variables had outer loadings higher than 0.6, which satisfies the theoretical requirement of [64], and confirms the scales' content validity.

**Table 3.** Confirmatory Factor Analysis (CFA).

| Latent Variables | Items | Loadings | Cronbach's Alpha | Rho_A | Composite Reliability | AVE |
|---|---|---|---|---|---|---|
| | Thresholds | | ≥0.6 | ≥0.7 | ≥0.7 | ≥0.5 |
| LEA | LEA1 | 0.634 | 0.826 | 0.848 | 0.877 | 0.589 |
| | LEA2 | 0.799 | | | | |
| | LEA3 | 0.810 | | | | |
| | LEA4 | 0.776 | | | | |
| | LEA5 | 0.820 | | | | |
| OC | OC1 | 0.753 | 0.864 | 0.867 | 0.898 | 0.594 |
| | OC2 | 0.800 | | | | |
| | OC3 | 0.733 | | | | |
| | OC4 | 0.802 | | | | |
| | OC5 | 0.749 | | | | |
| | OC6 | 0.786 | | | | |
| KM | KM1 | 0.835 | 0.850 | 0.854 | 0.899 | 0.690 |
| | KM2 | 0.821 | | | | |
| | KM3 | 0.823 | | | | |
| | KM4 | 0.843 | | | | |
| IC | IC1 | 0.843 | 0.855 | 0.864 | 0.902 | 0.697 |
| | IC2 | 0.825 | | | | |
| | IC3 | 0.883 | | | | |
| | IC4 | 0.787 | | | | |

We discuss the problem of discriminant validity using the parameter from the study of [65]. The square root of a construct's AVE must be greater than the association of any other construct. In addition, to evaluate whether any indicator loads strongly on other constructs, we analyzed all cross loadings of the indicators [57,65]. As presented in Table 4, all indicators satisfy the aforementioned criteria.

**Table 4.** Fornell–Larcker criterion.

| | Innovation Capability | Knowledge Management | Leadership Styles | Organizational Culture |
|---|---|---|---|---|
| Innovation capability | 0.835 | | | |
| Knowledge management | 0.693 | 0.831 | | |
| Leadership styles | 0.451 | 0.540 | 0.768 | |
| Organizational culture | 0.684 | 0.761 | 0.602 | 0.771 |

We also calculated for Heterotrait–Monotrait ratio (HTMT) to test for discriminant validity. This ratio depicts the mean value of all item correlations across constructs relative to the mean of the average correlations for items measuring the same construct. HTMT values are compared to a threshold of 0.85 [66,67]. If the HTMT value is higher than this threshold, it can be concluded that the model lacks discriminant validity. Since all the indicators obtained from our research were below 0.85, as reported in Table 5, we can conclude that there is high discriminant validity between the variables.

**Table 5.** Heterotrait–Monotrait Ratio (HTMT).

|  | Innovation Capability | Knowledge Management | Leadership Styles | Organizational Culture |
|---|---|---|---|---|
| Innovation capability |  |  |  |  |
| Knowledge management | 0.803 |  |  |  |
| Leadership styles | 0.520 | 0.625 |  |  |
| Organizational culture | 0.787 | 0.874 | 0.689 |  |

### 4.2. Structural Equation Model

After testing convergent and discriminant validity, the following step is to test the proposed hypotheses. Particularly, the structural model's predictive strength is determined by the $R^2$ values of the endogenous constructs [52,64]. In addition, $R^2$ values for endogenous constructs are considered substantial if $R^2 \geq 0.26$, moderate if $R^2 \geq 0.13$, and weak if $R^2 \geq 0.02$ [68]. As can be seen from Table 6, $R^2$ values of the endogenous constructs lie within the substantial range. This means that leadership styles, organizational culture and knowledge management factors explain 48.1% of the variance in innovation capability. It is worth noting that 59% of knowledge management variance is justified by leadership styles and organizational culture. Whereas, leadership styles account for 36.2% of variance in organizational culture construct.

**Table 6.** R-square of endogenous constructs.

| Construct | R-square | Result |
|---|---|---|
| Innovation capability | 0.481 | Substantial |
| Knowledge management | 0.590 | Substantial |
| Organizational culture | 0.362 | Substantial |

The following section analyzes the path coefficients of the targeted relationships. In order to test the hypotheses, we assessed path coefficients and their respective significant values. We employed a bootstrapping procedure to calculate significance values for all paths [57]. Figure 2 illustrated the results of hypotheses testing. All four hypotheses proposed were confirmed significant as their *p*-values were less than 0.1.

### 4.3. Discussions

As can be seen from the results, the relationship between leadership styles and organizational culture was confirmed to be positively significant (*p* < 0.001). Studies from [31,35,37] also provide the same findings. The results from our studies imply that supportive and participative management behaviors are important to create a collaborative, trusting and learning incentive business culture.

The relationship between organizational culture and knowledge management was also confirmed to be positively significant (*p* < 0.001), as Hypothesis 2 predicted. In other words, our results confirm that knowledge creation is associated with cultural factors such as collaboration, trust and learning. This finding is supported by studies from [4,18,45]. Moreover, a business environment in which collaboration, mutual trust and learning are encouraged can strongly enhance knowledge sharing, transfer and processes within the organization [4]. The research by [69] also claims that the ability to gain, organize,

and distribute knowledge is strongly correlated to the quality of the decision-making process. It can be interpreted that, when there is a high level of cooperation among group members, information and knowledge sharing can be accelerated as they have strong mutual trust. Overall, this finding shows strong support for the fact that develops a strong organizational culture plays crucial roles in creating sustainable, long-term as well as competitive advantages for firms operating in the high-tech industry [46].

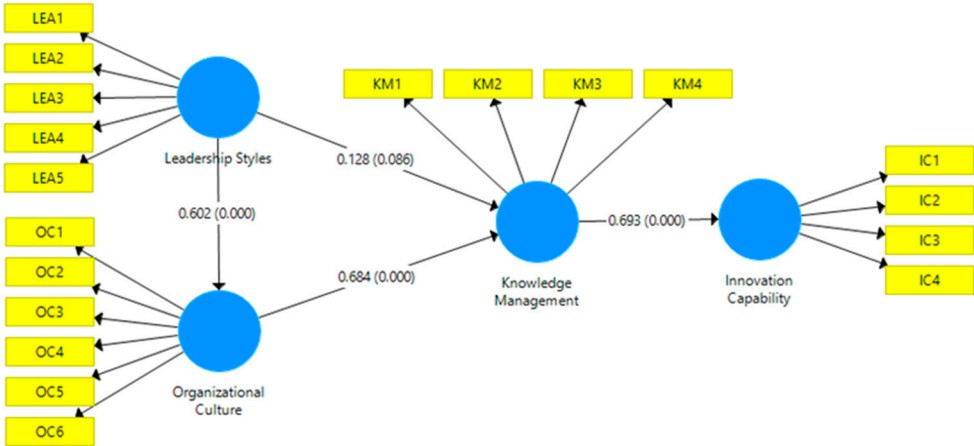

**Figure 2.** PLS-SEM results (*p*-values in brackets).

Table 7 showed that Hypothesis 3 was accepted (*p* < 0.1), meaning that there is a positive correlation between leadership styles and knowledge management. Our findings confirm that the beliefs, values, and actions of leaders are beneficial in creating an environment that fosters both cooperation and competition between departments. Based on social network theory, a network including complementary cooperative and competitive ties offer access to a vast market of information and eventually facilitates the sharing of best practices and relevant knowledge [55,57]. A considerate and participatory leadership behavior facilitates open exchange, mutual respect, and personal trust among organizational members and enhances employees' conflict solving skills [16]. The ability to identify, assimilate, transform and deploy information through intense interdepartmental interactions can be facilitated by a stimulating work atmosphere arising from the cooperative behavior of leaders. The findings were also consistent with the results of [16].

**Table 7.** PLS-SEM path coefficients.

| Hypotheses | Path Coefficients | *p*_Values | Results |
|---|---|---|---|
| H1: Leadership styles → Organizational culture | 0.602 | 0.000 *** | Accepted |
| H2: Organizational culture → Knowledge management | 0.684 | 0.000 *** | Accepted |
| H3: Leadership styles → Knowledge management | 0.128 | 0.088 * | Accepted |
| H4: Knowledge management → Innovation capability | 0.693 | 0.000 *** | Accepted |

Note: * *p* < 0.10, *** *p* < 0.01.

As predicted in Hypothesis 4, the relationship between knowledge management and innovation capability was found to be positively significant (*p* < 0.001). Previous studies also confirm the same finding. For instance, the study from [70] claims that effective knowledge management is essential to enhance innovation within an organization, especially for IT firms. Similarly, research by [51] on IT firms in China confirms that knowledge acquisition indeed positively influences innovation capability. When knowledge management practices are conducted efficiently, members of the organization are more willing to open to innovation applications. In other words, acquiring, applying, and sharing knowledge between the functional departments via internal networks can promote eagerness of or-

ganizational members to participate in creative activities, which can eventually lead to innovation enhancement [16,51].

Overall, the findings of our results imply that firms with a high level of collaboration, as well as significant platforms for sharing knowledge among employees and departments within an organization are more likely to encourage an open innovative environment [4,20]. In addition, knowledge management creates efficient information systems for sharing internal and external knowledge within firms locating high-tech zones. This can be a critical factor for innovation capability enhancement, as well as the successful determinant of the open innovation strategy implementation [22,25].

Regardless, this research still remains some limitations in the interpretation of organizational culture in dynamic concept. There is a lack of in-depth development of open innovation theory regarding the Vietnamese economy. This can also be considered as a major direction for further research.

## 5. Conclusions

In this research, a theoretical framework is proposed to identify empirical links among organizational culture, knowledge management and innovation capability. This study identified successful knowledge management resulted from solid business culture as one of the essential requirements of Vietnam IT firms' innovation capability. The research addressed this through three main objectives. First of all, we intend to identify and evaluate the attributes of leadership styles in the formation of organizational culture in Vietnamese high technology companies. Second, in the sense of the Vietnamese IT industry, we want to provide empirical evidence of the main characteristics of organizational culture that contribute to successful knowledge management practices. Third, we explore the relationship between knowledge management and innovation capability and investigate whether companies and organizations operating in high technology parks in Vietnam can generate competitive advantages while taking advantage of an efficient knowledge management implementation and improving the capacity of technology.

After carefully and thoroughly revising the questionnaire built upon previous studies, a survey was carried out to collect data from Vietnamese IT firms' representatives. A total of 182 responses were obtained, which were then used to conduct the quantitative analysis via PLS-SEM approach. Firstly, the current study finds that leadership styles have a significant impact on organizational culture. Secondly, our results indicate that organizational culture is strongly associated with knowledge management practices such as creation, storing and transferring information among departments. In a business environment where there is a high level of mutual trust, collaboration and learning, the knowledge exchange activities are more likely to occur frequently and effectively as boundaries between functional departments are reduced and the openness among organization members is increased. In addition, supportive and participative leader behaviors were empirically found to be strongly correlated with knowledge management. Finally, our research provides evidence for a significant relationship between innovation capability and knowledge management that is resulted from the established strong culture.

From a practical point of view, the relationship among organizational culture, knowledge management and innovation capability can provide useful insights for managers regarding developing a strong culture, promote knowledge management practices effectively and eventually enhance the whole organization's innovation capability. The creativity of an organization, especially one in the high-tech industry, provides a key to the understanding of organization effectiveness, growth and survival. Our model incorporated innovation capability because it is the seed of all innovation and represents the firms' ability to transform knowledge into business value. Neglecting the relationship between innovation capability and knowledge management may undermine a business environment. Shaping cultural factors is crucial for a firm's ability to manage knowledge effectively as a trust-based and open-minded business environment strongly encourages the organizations' members to participate in knowledge exchange activities via networking

relationships. Particularly, QTSC has established a Chief Executive Officer Club to build a business network and share business information among companies at the science park. Consequently, managers at QTSC are able to collaborate on maintaining competitive advantages through sharing knowledge and improving professional skills. In addition, as interpreted from our findings, leadership styles not only significantly relate to organizational culture, but also has a positive direct impact on knowledge management. Thus, it is recommended that managers and team leaders in IT firms should exhibit supportive and participative behaviors toward their subordinates to form a knowledge-nurtured culture. Specifically, managers at the science parks should organize more outdoor events such as picnics, festivals and sporting activities for employees getting together, having fun with their team, and sharing ideas.

**Author Contributions:** L.L. conducted in-depth interviews to complete the survey questionnaire, gave much research advice. P.N. developed, wrote the methodology, discussions, conclusions and revised all sections. N.L. wrote literature review and outlined the introduction section. K.T. wrote results and conclusion. P.N. is responsible for writing—review and editing, visualization, supervision, project administration, and funding acquisition. All authors have read and agreed to the published version of the manuscript.

**Funding:** This research is funded by Science and Technology Development Fund, Department of Science and Technology, Ho Chi Minh City, Vietnam under the contract number 03/2020/HĐ-QPTKHCN.

**Institutional Review Board Statement:** Not applicable.

**Informed Consent Statement:** Not applicable.

**Data Availability Statement:** Data is available on request.

**Conflicts of Interest:** The authors declare no conflict of interest. The funders had no role in the design of the study; in the collection, analyses, or interpretation of data; in the writing of the manuscript, or in the decision to publish the results.

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
