# Peer review of "The Relation among Organizational Culture, Knowledge Management, and Innovation Capability: Its Implication for Open Innovation"

_2199-8531, doi:10.3390/joitmc7010066_

Round 1
Reviewer 1 Report
Some key insights from the results from within specific demographic groups of the survey can further enhance the paper.
Author Response
Dear Reviewer;
Thanks for your suggestion, we have attempted to conduct a group analysis. However, the results have no significant difference. Therefore, we did not present in the paper.
We keep your suggestion for the future study.
Reviewer 2 Report
The review of this article was done in an open way, as the authors' identity was on the first page of the text and on the contributors' section. The entire text of the article is very well written. The objectives are clear and relevant. The authors were concerned with explaining their objectives in great detail. The study focuses Vietnam to investigate the relationship between organizational culture, knowledge management and innovation capability, which is a topic of great interest. Section 2 is very well written and built. Personally, I don't like phrases, such as: According to [5]. I would prefer to remove these expressions, leaving only the number at the end of the sentence. Or mention the cited authors, but leave the number to indicate the final reference. Section 3 is explained in a very detailed and rigorous way. I have nothing to report as negative. I think it would be more correct to separate the methods from the results. I particularly liked that the questionnaire was well-founded in the literature. Section 4 is also good. Conclusions are fine. 66 references and no evidences of self-plagiarism are a good quality indicator.
Author Response
Dear Reviewer;
Thanks for your comments and suggestions.
We have carefully reviewed the comments and have revised the manuscript accordingly. Our point-to-point responses are given in the table below. All revisions in the manuscript are clearly highlighted with yellow color.

Reviewer 3 Report
This is a reasonable paper, but the conclusions are unremarkable. They simply reveal what existing literature suggests and does not provide particularly surprising or insightful conclusion. It is difficult to attract attention unless the paper offers more substantial and new arguments.
Author Response
Dear Reviewer;
We have carefully reviewed the comments and have revised the manuscript accordingly. Our point-to-point responses are given in the table below. All revisions in the manuscript are clearly highlighted with yellow color.
Best regards,
Corresponding author

Round 2
Reviewer 3 Report
I received only one page saying that my concerns would be addressed. There were not other comments attached.
Author Response
Dear Reviewer;
I have uploaded the revised paper to the journal website. Therefore, I thought that you can access and see it.
I am sending it as the attached file.
Tones of thanks.
Best regards,
